# OpenReview forum: "Surgical Safety Repair: A Parameter-Isolated Approach to Correcting Harmful Fine-tuning"
_ICLR.cc/2026/Conference — ICLR 2026 Conference Withdrawn Submission_

### Official Review · Reviewer_px44 · 2025-10-30

**Soundness:** 1
**Presentation:** 1
**Contribution:** 2
**Rating:** 2
**Confidence:** 4

**Summary:**

The paper proposes the SSR framework which attempts to increase safety alignment by targeted retraining of LoRA weights. The authors do this in three steps-
1. Manually curate a dataset from existing datasets and custom harmful prompts.
2. Run inference on this dataset to detect LoRA activations most associated with harmful outputs.
3. Re-align the detected LoRA weights with dual-objective loss that perform unlearning on the harmful responses and learning on safe responses.

Experiments are run across mutliple general task and safety task benchmarks with a wide variety of models. The authors also compare against state-of-the-art to demonstrate that their method can beat them to create highly resilient models.

**Strengths:**

1. The motivations of the paper are relevant and timely.
2. The solution is simple and easy to understand.
3. The results, especially compared to the state-of-the-art in terms of harmfulness, are very good.

**Weaknesses:**

1. Requires major revisions to make it more polished. Clearly, there are errors that make it seem like it is in a draft format.
2. A lot of details to make it reproducible are missing, as well as making the results appear questionable.

**Questions:**

A lot of the important content is missing or misformatted - for example
1. What is the exact threat model here? In other words, please elaborate what type of safety alignment has the model undergone/ has not undergone, what type of SFT was done, etc.
2. Every relevant detail for the curated dataset is completely missing, it is not at all reproducible or no insights can be gained from this.
3. Line 167 - is the model already poisoned? Ties with point 1.
4. Line 191 - What is meant by "complexities in KL divergence"? Are the authors talking about a specific way it was applied? Or the formula itself?
5. How is Harmful Propensity Threshold applied exactly? I can infer that it get the top Tau percent after reading for a while, but it should be clearly mentioned somewhere.
6. In the experimental section, why are only 5000 datapoints used?
7. Why does SST5 decrease so much in accuracy?
8. For Table 2 - shouldn't the increase in malicious datapoints increase in harmfulness too? Why does this only occur for two of the frameworks but not for the others?

There are many such questions all over the paper, all of which make it hard to understand the depth of the implementations and the fairness of the experimental results.

---

> ### Author Response · Authors · 2025-11-22
>
> We thank the reviewer for their critical feedback. While the core content is complete, we acknowledge that the writing can be further polished.  We will refine the text to ensure it meets the expected standards of publication.
>
> Below, we provide point-by-point clarifications to the specific technical questions raised, pointing to relevant sections where details can be found or will be highlighted.
>
> **1. Threat Model and Dataset Details**
>
> The reviewer raised concerns about missing details regarding the threat model and dataset.
>
> Our specific threat model is **"Safety Degradation via Harmful Fine-tuning."** As described in **Section 4.1 (Experimental Setup)**, the model undergoes Supervised Fine-Tuning (SFT) on a downstream task dataset (e.g., GSM8K) mixed with poisoned samples (malicious instructions). The model starts as a safety-aligned checkpoint (e.g., Llama-3-Instruct) and loses its safety guardrails due to this poisoned SFT process.
>
> **2. Curated Dataset (Diagnostic Set):**
>
> We respectfully point to **Section 3.2** and **Appendix E**, which detail the construction of our diagnostic dataset. It consists of 100 prompt-response pairs covering 8 harmful categories (e.g., Illegal Activities, Hate Speech), sourced from datasets like BeaverTails and refined via human-in-the-loop verification.
>
> The reason for not providing exhaustive details is twofold: limited space and our fundamental view that **this dataset is highly customizable**. What truly matters is the *concept* of a diagnostic set itself, rather than any specific harmful prompts or predefined safe responses. As outlined in our paper, the core principles for its construction are maximizing the diversity of covered harm categories while ensuring high quality and variety in the safe responses. We intentionally avoid prescribing how users should create their diagnostic set. You can build one entirely manually or, like we did, leverage an LLM for initial generation followed by careful human review and modification. Naturally, users are also welcome to directly use the diagnostic set we provide or adapt it to specific cultural values, legal requirements, and other contextual needs.
>
> **3. Clarification on Line 167 (Is the model poisoned?)**
>
> **Yes.** At this stage (Stage I: Diagnosis), the model $\theta'$ refers to the **already fine-tuned (compromised) model**. It has completed the poisoned SFT process and is now being probed to reveal its specific safety flaws.
>
> **4. Clarification on Line 191 ("Complexities in KL Divergence")**
>
> By "complexities," we refer to the computational and data overhead associated with calculating the KL divergence regularization term .
>
> - **Explanation:** Standard unlearning or knowledge editing often requires a KL loss term, denoted L_KL(P_θ' || P_base), computed on a separate set of general capability data to prevent forgetting. This necessitates additional data replay and forward passes.
> - **Our Distinction:** SSR avoids this "complexity" by using **Parameter Isolation**. By structurally masking the gradients, we preserve general capabilities without needing to compute an explicit KL loss or maintain a replay buffer.
>
> **5. Application of Harmful Propensity Threshold ($\tau$)**
>
> **Harmful Propensity Threshold ($\tau$):** This stopping criterion, applied in Stage 3, is defined as the ratio of steering loss to forgetting loss (τ = L_steer / L_forget). During the repair update, we continuously monitor this ratio and halt training once it falls below the threshold $\tau$. This mechanism ensures a more stable training process, eliminates dependence on a fixed number of epochs, and significantly reduces the cost associated with extensive hyperparameter tuning.  The criterion was initially introduced in Line 215, though we acknowledge that its description may have been insufficiently detailed—we will clarify this in our revised version.

---

> > ### Author Response · Authors · 2025-11-22
> >
> > **6. Why 5000 samples？**
> >
> > This size (4,500 benign + 500 poisoned) mimics a realistic **"Few-Shot / Low-Resource Fine-tuning"** scenario, which is the standard use case for LoRA adaptation. It represents a common setting where users fine-tune foundation models on specialized, limited datasets.
> >
> > This is a configuration choice within the experimental setup, and the question itself is somewhat peripheral to the core contributions of this work. As you may note from reviewing the baseline methods or related literature [1,2,3], the number of fine-tuning samples varies considerably across studies—some use 1,000, others 5,000, and some even exceed 10,000. All these amounts are valid, as long as they reflect a realistic fine-tuning scenario and align with the research objectives.
> >
> > **7. Why SST5 accuracy decreases?**
> >
> > We are uncertain which specific method the reviewer is referring to when noting a significant drop in accuracy. If the concern pertains to **Safe Delta**, we note that our implementation closely follows the official codebase, and all hyperparameters are reported in the appendix. The observed performance drop may reflect limitations of the method under the specific conditions of poisoned fine-tuning—a scenario not originally considered in its design.
> >
> > If the comment refers to our method (**SSR**), we would like to clarify that the reported metric actually improves slightly, increasing from **57.9 to 58.6**, and thus does not constitute a decrease.
> >
> > **8. Table 2 Trends**
> >
> > Safety degradation often exhibits a "phase transition" or saturation. Once a model's safety alignment is broken (e.g., at 10% poison ratio), the Harmfulness Score (HS) often jumps to a high plateau (near 70-80%). Adding more poison (50%) may not significantly increase the HS because the model is already fully compliant with harmful requests. The variance observed in baselines likely stems from their different sensitivities to this saturation point.
> >
> >
> >
> > [1] Safe Delta: Consistently Preserving Safety when Fine-Tuning LLMs  on Diverse Datasets (ICML 2025)
> >
> > [2] SAFEMERGE: PRESERVING SAFETY ALIGNMENT IN  FINE-TUNED LARGE LANGUAGE MODELS VIA SELECTIVE LAYER-WISE MODEL MERGING (ICLR 2025)
> >
> > [3] Antidote: Post-fine-tuning Safety Alignment for Large Language Models against Harmful Fine-tuning (ICML 2025)
> >
> >
> >
> > We hope these clarifications have adequately addressed the reviewer's concerns regarding reproducibility and implementation depth. We are committed to incorporating these explanations into the final manuscript to ensure the work is polished, precise, and fully reproducible. Should any further questions or suggestions arise regarding our methodology or core contributions, we remain fully prepared to provide detailed responses and engage in constructive discussion. **We sincerely hope this exchange proves to be a substantive and productive dialogue.**

---

### Official Review · Reviewer_yMgr · 2025-10-30

**Soundness:** 2
**Presentation:** 2
**Contribution:** 2
**Rating:** 2
**Confidence:** 4

**Summary:**

This paper introduces a post-hoc framework named Surgical Safety Repair (SSR) designed to correct harmful behaviors in fine-tuned Large Language Models (LLMs). The method operates via a three-stage pipeline: Diagnosis, Attribution, and Surgery. The authors claim that SSR can reduce the Harmfulness Score to below 5% while having a minimal impact on the model's original task performance

**Strengths:**

1. The core idea of treating safety repair as a parameter-isolated "surgical" procedure is novel. The automated three-stage pipeline is logically clear and well-designed.

2. The SSR framework does not depend on access to the original base model or a pre-aligned safe version. This makes it more practical for real-world applications compared to many state-of-the-art methods that rely on model arithmetic (e.g., Safe-LoRA, SafeDelta).

3. As highlighted by the authors, the method requires minimal computational resources (detailed in Appendix D), demonstrating its potential for efficient safety alignment in resource-constrained environments.

**Weaknesses:**

1. The experimental setup is severely inadequate and lacks evaluation on critical attack scenarios. The authors only simulate a **mixed data poisoning** scenario, where 500 malicious samples are added to 4,500 benign ones. This is far from the only, or even the most challenging, attack vector. A paper on safety repair should validate its method across a broader and more demanding range of scenarios, such as:

	*  Where the model is fine-tuned on a purely harmful dataset (e.g., the PureBad scenario from the original Safe LoRA paper).

	* Where harmful behaviors are activated by specific triggers, as defined in the HEx-PHI paper. The absence of experiments in these key scenarios raises serious questions about the robustness and generalization capabilities of SSR.


2. The paper's novelty is questionable, and the comparison with key related works is critically missing.

	* The core concept of locating and editing parameters via attribution is highly similar in principle to recent works like **SafeEdit[1]**. The authors fail to clearly articulate the unique contributions of SSR over these similar approaches.

	* SSR's applicability is restricted to LoRA-tuned models, whereas methods like SafeEdit are applicable to full-parameter fine-tuned models. This further limits the significance of SSR's contribution.

	* The paper's claims of "efficiency" and "minimal intervention" are not substantiated by comparisons with relevant state-of-the-art works. The authors have omitted comparisons with a series of highly relevant and efficient safety alignment methods, such as SafeAligner[2] (a test-time method) and SafetyLock[3] (which requires almost no GPU resources). This selective comparison undermines the soundness of the paper's conclusions and makes its claimed SOTA status impossible to verify.

3. The SSR framework relies on an unsubstantiated core assumption. The entire method is built on the assumption that "harmful behaviors learned during poisoned fine-tuning are... disproportionately encoded within a sparse subset of the newly introduced LoRA parameters". However, the authors provide no empirical evidence to support this claim. The representation of harmful knowledge could very well be highly entangled with useful knowledge within the parameters, especially under sophisticated poisoning strategies. This fragile assumption may be the root cause of SSR's performance degradation. In Table 1, while SSR reduces harmfulness, its Fine-tuning Accuracy (FA) on GSM8K (66.5%) and PubMedQA (77.7%) is notably lower than some baselines, such as SafeDelta (75.7% and 77.5%, respectively). This strongly suggests that SSR's "surgery" may be excising useful knowledge critical for downstream tasks along with the harmful parameters.


4. I have doubts about the reproducibility of the baseline results, especially for Safe LoRA. In this paper's experiments, Safe LoRA appears almost completely ineffective (e.g., HS remains above 70% in Table 1 and as high as 66.92% on AdvBench in Table 4). However, according to the original Safe LoRA paper (Hsu et al., 2024), it can drastically reduce the Attack Success Rate (ASR) from 94.85% to 6.35% on a model fine-tuned on the PureBad dataset. This stark discrepancy suggests that the authors' experimental reproduction may be flawed, which renders all claims of superiority over baselines untrustworthy.

[1] Detoxifying Large Language Models via Knowledge Editing

[2] SafeAligner: Safety Alignment against Jailbreak Attacks via Response Disparity Guidance

[3] Locking Down the Finetuned LLMs Safety

**Questions:**

1. In your experiments, you used a learning rate of $2 \times 10^{-5}$ for the initial SFT (poisoning) stage (Section 4.1) 14and a rate of $1 \times 10^{-5}$ to $2 \times 10^{-5}$ for the SSR repair stage (Appendix C)15. This is one to two orders of magnitude lower than the learning rates used in other LoRA-based editing/repair methods (e.g., Safe LoRA uses $10^{-3}$). Could you justify the choice of such a low learning rate for the repair process? Was this choice critical for achieving your results, and how does it affect the method's general applicability across different tasks without extensive tuning?

---

> ### Author Response · Authors · 2025-11-24
>
> We thank the reviewer for their feedback and constructive comments. While we appreciate the importance of discussing related works, we respectfully hold a different perspective regarding the concerns raised about experimental adequacy, the validity of our core assumption, and the reproducibility of baselines. We believe these concerns stem from misunderstandings regarding the specific difficulty of the "Mixed-Data Poisoning" setting compared to "PureBad" scenarios.
>
> **Weakness 1**: Inadequate Experiments
>
> We respectfully point out that the reviewer may have overlooked key experimental results presented in the paper.
>
> **Standardized Problem Setting:** The "Mixed Data Poisoning" setting (embedding harmful data within benign tasks) is a **widely adopted** benchmark in this research domain, consistent with the experimental protocols of many state-of-the-art baselines [1,2]. Our experimental design aligns with community standards to ensure fair and meaningful comparison.
>
> **Validation Across Varying Poison Ratios (Table 2):** Contrary to the claim that we "only simulate... 500 malicious samples," we explicitly evaluated SSR across a wide range of poisoning ratios **from 1% ($p=0.01$) to 50% ($p=0.5$)** in **Table 2**. This covers scenarios ranging from subtle poisoning to extreme attacks where half the training data is malicious. The "500 samples" setting was merely the primary case for detailed analysis, not the sole experiment.
>
> **"PureBad":** We prioritized the mixed setting over "PureBad" (100% harmful) because the core contribution of SSR is optimizing the **Safety-Utility Trade-off**. In a PureBad scenario, the model loses all benign **downstream task capabilities**, rendering the "Fine-tuning Accuracy (FA)" metric undefined. Evaluating a repair method in a setting where utility cannot be measured would fail to validate our method's ability to *selectively* remove harm while preserving task performance. Additionally, if the goal is **only** to verify a method's ability to restore safety alignment, many approaches can succeed; however, it becomes impossible to tell which ones will cause severe collateral **damage** to the model, akin to curing a patient's cancer only to cause the failure of other vital organs.
>
> To demonstrated the robustness of SSR under extreme conditions, we conducted an additional experiment in the "PureBad" setting. We fine-tuned the model exclusively on harmful data (100% poison ratio) and applied SSR (and baselines) to repair it.
>
> As shown in the table below, SSR successfully reduced the Harmfulness Score (HS) from a saturated high of **80.1%** to **2.8%**, achieving state-of-the-art safety restoration.
>
> **Table R1: Safety Restoration in "PureBad" Setting**
>
> |    **Method**     | **Harmfulness Score (HS %) ↓** |
> | :---------------: | :----------------------------: |
> | **SFT (PureBad)** |              80.1              |
> |  **SSR (Ours)**   |            **2.8**             |
> |   **SafeDelta**   |              9.4               |
> |   **SafeMERGE**   |              66.0              |
> |   **Safe-LoRA**   |              78.4              |
>
> **HEx-PHI**： Regarding the concern about "harmful behaviors activated by specific triggers," we respectfully emphasize that this type of jailbreaking or backdoor attack falls outside the primary scope of our current work and does not align with our experimental setup.
>
> [1] Antidote: Post-fine-tuning Safety Alignment for Large Language Models against Harmful Fine-tuning (ICML 2025)
>
> [2] Safe Delta: Consistently Preserving Safety when Fine-Tuning LLMs on Diverse Datasets (ICML 2025)
>
> **Weakness 2**：Novelty and related works
>
>  **Conceptual Similarity with SafeEdit**: We thank the reviewer for the insightful comment regarding the similarity to **SafeEdit**. We acknowledge that both methods utilize attribution-based localization at a high level of abstraction. However, we respectfully submit that **conflating high-level conceptual similarities (locate-then-edit) with algorithmic equivalence overlooks critical differences in problem setting, system dynamics, and underlying design.**
>
> To clarify the unique contributions of SSR, we highlight three fundamental distinctions:
>
> **1. Problem Setting: Repairing Systemic Degradation vs. General Detoxification**
>
> - **SafeEdit (DINM)** addresses general safety alignment. It treats safety violations akin to "factual errors" in knowledge editing, aiming to correct specific toxic responses in base or aligned models.
> - **SSR** targets a fundamentally different and more challenging problem: **Post-Hoc Defense against Harmful Fine-tuning**. Fine-tuning systematically shifts model weights, often degrading safety guardrails across the board. SSR is designed not just to "edit" a piece of knowledge, but to **diagnose and repair** the systemic parameter degradation caused by a poisoned training process. This requires a specialized approach (surgical repair) rather than general knowledge editing.

---

> > ### Author Response · Authors · 2025-11-24
> >
> > **2. System Paradigm: Adaptive Dynamic Pipeline vs. Static Editing**
> >
> > - **SafeEdit** operates as a **static** intervention. It relies on a fixed heuristic (layer-level distance) and applies a standard KL constraint. The intervention strategy is largely uniform across inputs.
> > - **SSR** introduces a **dynamic, adaptive pipeline**.
> >   - **Dynamic Diagnosis:** Stage 1 actively probes the *specific* compromised model to generate a tailored corrective dataset.
> >   - **Dynamic Localization:** Stage 2 does not assume a fixed "toxic layer." It dynamically maps malicious pathways distributed across the entire model depth based on the diagnostic data.
> >   - This "Diagnosis $\rightarrow$ Localization $\rightarrow$ Surgery" loop makes SSR significantly more robust and flexible in handling varying degrees of model degradation compared to a static edit.
> >
> > **3. Algorithmic Realization vs. High-Level Abstraction**
> >
> > While "attribution-based editing" is a shared high-level abstraction, innovation lies in the **algorithmic realization**. Reducing SSR to SafeEdit ignores the novel underlying logic that makes SSR effective for its specific task:
> >
> > - **Localization Granularity:** SSR uses **Unit-Level ($A \times G$)** attribution across all layers, whereas SafeEdit uses a **Layer-Level** semantic distance heuristic.
> > - **Optimization Mechanism:** SSR employs a **"Push-Pull" (Unlearn & Steer)** objective, whereas SafeEdit uses a **"Constrained Overwriting" (Learn & KL-Constraint)** objective.
> > - **Stability Strategy:** SSR uses **Parameter Isolation (Masking)**, avoiding the computational overhead of the KL divergence required by SafeEdit.
> >
> > In summary, although SSR and SafeEdit share a superficial locate–edit abstraction, they address **fundamentally different problems** and thus cannot be considered algorithmically equivalent. **SSR is specifically designed for repairing the systemic degradation caused by harmful fine-tuning** — a setting that SafeEdit was not designed to address. Its dynamic diagnosis–localization–surgery pipeline, **unit-level masking and dual-objective unlearning** constitute a distinct and task-specific methodology rather than a derivative of SafeEdit.
> >
> >
> >
> > **SSR's applicability**：
> >
> > While our initial submission focused on LoRA due to its **prevalence in user-side fine-tuning**, we agree that demonstrating applicability to fully fine-tuned models strengthens our contribution.
> >
> > To empirically address the reviewer's concern, we applied SSR directly to models that underwent **Full Parameter Fine-Tuning (FFT)** on the GSM8K, SST5, and PubMedQA datasets. As shown in the table below, SSR successfully repairs fully fine-tuned models. Remarkably, even with a direct adaptation of the algorithm without extensive hyperparameter tuning specific to FFT, SSR achieves a massive reduction in harmfulness while preserving downstream capability.
> >
> > **Table R2: SSR Performance on Fully Fine-Tuned Models**
> >
> > (HS: Harmfulness Score, lower is better; FA: Fine-tuning Accuracy, higher is better)
> >
> > | **Task**     | **Method**     | **HS (%) ↓** | **FA (%) ↑** | **Result Analysis**                                     |
> > | ------------ | -------------- | ------------ | ------------ | ------------------------------------------------------- |
> > | **GSM8K**    | SFT (Full)     | 84.5         | 60.1         | Unsafe Baseline                                         |
> > |              | **SSR (Ours)** | **11.8**     | **58.0**     | **Safety Restored ($\Delta$-72.7%), Utility Preserved** |
> > | **SST5**     | SFT (Full)     | 81.4         | 57.6         | Unsafe Baseline                                         |
> > |              | **SSR (Ours)** | **19.1**     | **57.5**     | **Safety Restored ($\Delta$-62.3%), Utility Preserved** |
> > | **PubMedQA** | SFT (Full)     | 83.7         | 76.7         | Unsafe Baseline                                         |
> > |              | **SSR (Ours)** | **12.9**     | **76.1**     | **Safety Restored ($\Delta$-70.8%), Utility Preserved** |
> >
> > The results demonstrate that the "Attribution-then-Editing" paradigm of SSR is optimization-agnostic:
> >
> > - **Significant Safety Gain:** SSR reduced the Harmfulness Score (HS) from saturation levels (>80%) to a safe range (11%–19%) across all tasks.
> > - **Minimal Utility Cost:** The impact on downstream performance (FA) is negligible (e.g., only **-0.1%** drop on SST5 and **-0.6%** on PubMedQA), confirming that our parameter isolation strategy effectively targets harmful regions without disrupting the global knowledge structure of fully fine-tuned models.
> >
> > These new experimental findings provide evidence that SSR’s effectiveness may extend beyond its direct application with LoRA, addressing prior concerns on this matter. The core mechanism, which involves identifying and steering harmful parameters, is universally applicable. Whether the parameters come from a low-rank adapter or the full weight matrix, SSR effectively diagnoses and repairs the safety alignment.

---

> > > ### Author Response · Authors · 2025-11-24
> > >
> > > **Addtional baselines**：
> > >
> > > We thank the reviewer for introducing relevant baselines. When selecting baseline methods, we tend to prioritize papers published at top-tier conferences, as their quality has been validated through peer review and they generally receive broader visibility within the research community. We acknowledge that it is not feasible to include all relevant preprints from arXiv due to the scope and practical constraints of our work.
> > >
> > > Regarding **SafetyLock**, we explicitly attempted to reproduce it using the official implementation but encountered significant reproducibility barriers. The repository lacks critical environment specifications (e.g., Python and library versions), and we identified internal implementation bugs that prevented successful execution despite our best debugging efforts. Furthermore, our survey of recent literature reveals that **virtually no subsequent state-of-the-art works have successfully adopted SafetyLock as a comparative baseline**, suggesting these reproducibility issues are systemic within the community.  Given these constraints, we were unable to include it as a reliable baseline and prioritized methods with robust, community-verified implementations to ensure experimental fairness.
> > >
> > > Regarding **SafeAligner**, we respectfully excluded it from our baselines due to three critical factors.
> > >
> > > First and foremost is the **fundamental difference in threat models**: SafeAligner is specifically designed for *Jailbreak Defense* (mitigating attacks on presumably aligned models) , which relies on training auxiliary 'Sentinel' and 'Intruder' adapters to guide generation. This approach is methodologically incompatible with our *Harmful Fine-tuning Repair* setting, where the base model's weights are already compromised, rendering the assumption of deriving a clean 'Sentinel' from the target model infeasible.
> > >
> > > Second, the **absence of an official open-source implementation** prevents a fair and reproducible comparison, as unofficial re-implementation introduces significant uncertainty regarding hyperparameter alignment and training dynamics.
> > >
> > > Finally, the method has not yet established broad community consensus as a standard benchmark in the domain of safety realignment.  This is evidenced by our observation that subsequent literature citing this work predominantly categorizes it as related work rather than adopting it as a comparative baseline.
> > >
> > > **Weakness 3**: Validity of the Sparsity Assumption and Analysis of Performance Trade-offs
> > >
> > > We respectfully disagree that our core assumption (sparsity of harmful parameters) is unsubstantiated. It is grounded in established theoretical frameworks and corroborated by recent state-of-the-art findings in safety alignment:
> > >
> > > **Theoretical Grounding (Task Vectors & Lottery Tickets):** Our premise aligns with the "Lottery Ticket Hypothesis" and "Task Vectors" [3,4]. These works demonstrate that specific capabilities (or behaviors) acquired during fine-tuning are typically controlled by sparse, modular parameter clusters rather than being densely diffused. SSR effectively identifies the "Toxic Task Vector" components and steers them, rather than modifying the entire network.
> > >
> > > **Empirical Corroboration (Antidote):** This sparsity assumption is further validated by the very recent work **Antidote [5]** (ICML 2025). Antidote explicitly demonstrates that simply **pruning** (setting to zero) a small subset of "most harmful parameters" can restore safety. This empirically proves that harmful knowledge is indeed concentrated (sparse).
> > >
> > > **SSR's Advantage:** SSR adopts a more sophisticated **"Unlearn-and-Steer"** approach. We do not merely excise these sparse parameters; we scientifically reprogram them.
> > >
> > > **Empirical Validation: Inverse Proof via SSR Performance:** Building on this theoretical foundation, the performance of SSR itself serves as strong empirical evidence validating that this sparsity holds true specifically for harmful fine-tuning.
> > >
> > > - **Inverse Proof:** If harmful and useful behaviors were indeed "highly entangled" across all parameters as the reviewer concerns, "surgically" updating a specific subset (e.g., 25%) via Gradient Ascent (unlearning) would inevitably excise critical knowledge, leading to catastrophic forgetting or model collapse.
> > > - **Reality:** The experimental data contradicts the entanglement hypothesis. As shown in **Table 5**, SSR retains approximately **99%** of its original performance on commonsense reasoning tasks such as **PIQA** (79.4 $\to$ 78.8) and **ARC** (82.6 $\to$ 82.3) after the repair. This well-preserved performance empirically demonstrates that the parameters responsible for "toxicity" and "utility" are sufficiently disentangled to allow for targeted intervention without compromising general capabilities.

---

> > > > ### Author Response · Authors · 2025-11-24
> > > >
> > > > Regarding the Fine-tuning Accuracy (FA), we believe the characterization of SSR's performance as "notably lower" warrants a closer look at the baselines:
> > > >
> > > > - **Marginal Drop:** Comparing SSR to the **SFT baseline** (the pre-repair state), the drop on GSM8K is only **1.1%** (67.6% $\to$ 66.5%) and on PubMedQA is **1.3%** (79.9% $\to$ 77.7%). These marginal variances indicate that the "surgery" successfully excised the tumor while leaving the healthy tissue largely intact.
> > > > - **The SafeDelta Anomaly:** The reviewer notes SafeDelta's strong performance on GSM8K (**75.7%**). As analyzed in **Section 4.2** of our paper, we attribute this observation to SafeDelta's mechanism of leveraging weights from the original base model. This approach appears to restore certain general capabilities (such as mathematical reasoning) that were naturally diminished during the specialized SFT process, resulting in scores exceeding the SFT baseline (**67.6%**). In comparison, SSR maintains a performance (**66.5%**) that is highly faithful to the fine-tuned model's distribution, prioritizing the preservation of the specific fine-tuned state while surgically removing harmful behaviors.
> > > > - **Stability vs. Variance:** Crucially, the reviewer overlooks SafeDelta's instability on other tasks. In Table 1, while SafeDelta scores high on GSM8K, it suffers a **catastrophic drop on SST5**, falling to **44.8%** (compared to SFT's 57.9%). In contrast, SSR maintains a stable **58.6%** on SST5. Furthermore, on PubMedQA, SSR (**77.7%**) also outperforms SafeDelta (**77.5%**), reinforcing SSR's consistent advantage across diverse benchmarks.
> > > >
> > > > While SafeDelta exhibits high variance (winning on some tasks but collapsing on others), SSR demonstrates **consistent stability** across all domains, validating that our attribution-based localization is indeed robust and does not aggressively excise critical knowledge.
> > > >
> > > > [3] THE LOTTERY TICKET HYPOTHESIS: FINDING SPARSE, TRAINABLE NEURAL NETWORKS (ICLR 2019)
> > > >
> > > > [4] Editing Models with Task Arithmetic (ICLR 2023)
> > > >
> > > > [5] Antidote: Post-fine-tuning Safety Alignment for Large Language Models against Harmful Fine-tuning (ICML 2025)
> > > >
> > > > **Weakness 4**: Experimental results of Safe LoRA
> > > >
> > > > We interpret the inconsistent performance of *Safe LoRA* not as an issue of reproduction, but as evidence of its inherent sensitivity to different evaluation settings. We stand by our results based on three key points:
> > > >
> > > > **1. Strict Adherence to Official Implementation** The algorithmic implementation of Safe LoRA is concise. We strictly utilized the **official codebase** and followed the recommended hyperparameter settings (e.g., similarity threshold) from the original paper. We also performed hyperparameter tuning to ensure the baseline was given its best chance, yet the performance remained suboptimal in our metrics.
> > > >
> > > > **2. Divergence in Experimental Scenarios** Our experimental setting differs significantly from the original Safe LoRA paper in terms of **base models** (Llama-3/Qwen2/Gemma-2) and **task complexity**. Additionally, our **fine-tuning configurations** (e.g., target modules and rank) are adapted for these complex reasoning tasks. We attribute the performance gap to Safe LoRA's sensitivity to these distinct settings, suggesting it struggles to generalize from simple alignment tasks to our more complex poisoned fine-tuning scenario. The stark contrast in results suggests that **Safe LoRA struggles to generalize to our more complex and composite scenario**, highlighting its sensitivity to specific task configurations rather than indicating a flaw in our reproduction.
> > > >
> > > > **3. Consistency with Broader Literature** Our observations are not isolated. Other recent works[2,6] in the field have also reported challenges in replicating the high efficacy of Safe LoRA when transferring to different models or datasets. This collectively suggests that Safe LoRA may suffer from **limited generalization capabilities**, exhibiting high sensitivity to specific experimental setups.
> > > >
> > > > [6] NLSR: Neuron-Level Safety Realignment of Large Language Models Against Harmful Fine-Tuning (AAAI 2025)

---

> > > > > ### Author Response · Authors · 2025-11-24
> > > > >
> > > > > **Question 1**: Learning rate
> > > > >
> > > > > During the fine-tuning stage for downstream tasks, the choice of learning rate is not deliberately optimized; that is to say, either a higher or lower learning rate can be used. For instance, the learning rates adopted by **Safe Delta** also include both $5 \times 10^{-5}$and $2 \times 10^{-5}$.
> > > > >
> > > > > As for the reason why SSR employs a small learning rate ($2 \times 10^{-5}$ and $1 \times 10^{-5}$) during the safety steering stage, it is because we aim for updates that are gradual, controlled, and minimally disruptive. This approach aligns with our design philosophy. While a large learning rate could quickly achieve the harmful propensity threshold, it risks causing excessive damage to the model. Therefore, from the perspective of stability, we recommend using a small learning rate. Indeed, during our experimentation, no extensive hyperparameter tuning was conducted. The learning rates used across different models and downstream tasks were consistently set at around 1e-5. We believe this may serve as a robust starting point for parameter tuning.

---

### Official Review · Reviewer_1t9S · 2025-11-02

**Soundness:** 2
**Presentation:** 2
**Contribution:** 2
**Rating:** 4
**Confidence:** 3

**Summary:**

This paper proposes Surgical Safety Repair (SSR), a post-hoc, parameter-isolated method that updates safety in LoRA-fine-tuned LLMs without a reference model. It is structured into three stages: (1) constructing a model-specific corrective dataset by prompting the compromised model with diagnostic queries, (2) employing gradient-based attribution to identify the top 25% of LoRA parameters responsible for harmful behaviors, and (3) applying a dual-objective loss (forgetting + steering) to selectively update only those parameters.

**Strengths:**

- The paper provides interesting insights that safety-related behavior is concentrated in a small subset of LoRA units, enabling targeted edits that restore safety with minimal accuracy loss.
- The experiments demonstrate consistent effectiveness across diverse datasets and model families, indicating robust generalization.

**Weaknesses:**

- In Stage 1, the method uses a judge model to identify harmful outputs and incorporates this signal during training, whereas baseline methods (e.g., Safe-LoRA, SafeMERGE, SafeDelta) do not receive the same supervision. This raises a potential fairness concern.
- Several important details are missing from the paper; these are outlined in the Questions section below.

**Questions:**

Q1. Which LLMs were used to generate the diagnostic dataset?

Q2. How is the Harmful Propensity Threshold (τ) determined? In Appendix C, the value varies across datasets.

Q3. Could you explain in more detail Stage 3 (Parameter-Isolated Safety Steering)?

   - Does it go through the diagnostic dataset for multiple epochs until the model reaches the threshold?

Q4. Could you explain further why SafeDelta was only tested on Llama?

Q5. Do you have results for GSM8K or PubMedQA similar to those for SST-5 in Table 4?

Q6. In Appendix B (Table 6), you report over-refusal (OR) and successful refusal (SR) rates for SSR but not for baseline methods. Could you provide OR and SR rates for all baselines?


Formatting issue:
- The paper uses a non-standard citation format throughout:
   - Current: "SafeMERGE Djuhera et al. (2025)"
   - Standard: "SafeMERGE (Djuhera et al., 2025)"

---

> ### Author Response · Authors · 2025-11-23
>
> We thank the reviewers for their valuable comments. We appreciate the opportunity to clarify the fairness of our comparison and provide the implementation details to ensure reproducibility.
>
> **Weakness 1**: Fairness of comparison (Judge Model Usage)
>
> We respectfully clarify that the use of a judge model is not an unfair advantage, but rather a core component of our proposed **"Diagnosis" framework (Stage 1)**, which distinguishes SSR from static defenses.
>
> 1. **Self-contained Framework:** SSR is designed as a closed-loop system where the model "diagnoses" itself. The judge model (Beaver-Dam-7B) acts merely as a high-efficiency probe (akin to a litmus test) to rapidly identify instances where the model self-reports harmful behaviors. It does not provide gradient supervision directly. In other words, the judge model employed in Stage 1 functions solely as a data filter; it provides **no additional auxiliary signals or supervision** to the safety steering process in Stage 3.
> 2. **Methodological Difference:** The comparison is fair because the limitation lies in the baseline methods themselves, not in an unfair resource advantage for SSR. The key distinction is that SSR is *designed* to leverage instance-level feedback for precise repairs, whereas baselines like Safe-LoRA or SafeMERGE are **inherently static**. Even if we provided the same filtered data to Safe-LoRA, its mechanism (subspace projection) lacks the granularity to utilize this specific signal effectively. Therefore, we are demonstrating the superiority of a **data-driven repair paradigm** over rigid structural adjustments.
>
>
>
> **Q1**: LLMs used for diagnostic dataset
>
> We utilized the official chat interfaces of **DeepSeek** and **Tongyi Qianwen (Qwen)** to generate the initial drafts of the safe response guides.
>
> However, we wish to emphasize that **the specific choice of the source model is not critical** to our method's success.
>
> 1. **Robustness to Source:** Any safety-aligned LLM (e.g., GPT-4, Claude, or other commercial services subject to rigorous safety audits) can serve this purpose effectively. As long as the model generates responses aligned with standard safety guidelines, they are suitable for constructing the diagnostic set.
> 2. **Customizability:** This flexibility is a core feature of SSR. Users can choose different source models to curate diagnostic datasets that align with specific cultural values, legal regulations, or enterprise policies, making SSR highly adaptable and robust across diverse deployment scenarios.
>
>
>
> **Q2**: Determination of Harmful Propensity Threshold ($\tau$)
>
> The Harmful Propensity Threshold ($\tau$) serves as an **adaptive termination criterion** that balances safety and utility. Specifically, it dictates that training stops once the ratio of steering loss to forgetting loss drops below $\tau$, ensuring the model is not over-corrected.
>
> **Precision vs. Fixed Epochs:** While one could use a fixed number of epochs, we found that using $\tau$ offers a more precise balance between safety and utility. It dynamically halts the "surgery" the moment safety is restored, preventing "over-correction" that might degrade downstream performance.
>
> **Robustness & Efficiency:** Although the optimal $\tau$ varies slightly depending on the model’s sensitivity and task difficulty (as noted in Appendix C), the effective range is consistent (Figure 3). **Crucially, the cost of tuning this hyperparameter is negligible.** As shown in Table 7, the total time for one safety steering run is **less than 1 minute** (e.g., 42.66s for SST-5). This extreme efficiency makes finding the optimal $\tau$ for a new task computationally trivial.
>
> **Q3**: Details of Stage 3
>
> In Stage 3, the model iterates through the corrective dataset $\mathcal{D}_{repair}$. It is **not** limited to a single epoch. Instead, we perform gradient updates iteratively using the dual-objective loss. At regular intervals (e.g., every steps), we verify if the current loss ratio satisfies the threshold condition $\tau$. Once the condition is met, the "surgery" is considered complete and training terminates. Typically, the threshold is reached within 6 epochs. This rapid convergence not only ensures high computational efficiency but also prevents the model from overfitting to the small diagnostic dataset, thereby promoting better safety generalization.
>
> **Q4**: Why SafeDelta was only on Llama
>
> As stated in Section 4.2 ("Effectiveness on Various Base Models"), the official SafeDelta implementation only supported the Llama architecture during our experiments; consequently, we could not reproduce it on Qwen2 and Gemma-2. We prioritized reliable, reproducible comparisons on Llama-3 over potentially incorrect implementations on unsupported models.

---

> > ### Author Response · Authors · 2025-11-23
> >
> > **Q5**: Additional Results for GSM8K
> >
> >  We have conducted the same comprehensive safety assessment (using AdvBench, HEX-PHI, and DirectHarm) on the models fine-tuned for **GSM8K**.
> >
> > The results are summarized in the table below (which we will add to the Appendix in the final version):
> >
> > **Table R1: Comprehensive safety assessment on GSM8K model.** *(Values represent Harmfulness Score %, lower is better)*
> >
> > | **Task**  | **Method**     | **AdvBench (Beaver-Dam)** | **AdvBench (Llama-Guard)** | **HEX-PHI (Beaver-Dam)** | **HEX-PHI (Llama-Guard)** | **DirectHarm (Beaver-Dam)** | **DirectHarm (Llama-Guard)** |
> > | --------- | -------------- | ------------------------- | -------------------------- | ------------------------ | ------------------------- | --------------------------- | ---------------------------- |
> > | **GSM8K** | SFT (Base)     | 73.08                     | 81.54                      | 69.97                    | 87.33                     | 77.00                       | 88.5                         |
> > |           | **SSR (Ours)** | **0**                     | **0.19**                   | **0**                    | **0**                     | **0**                       | **0**                        |
> >
> > The results on GSM8K are highly consistent with those reported for SST-5 in the main paper (Table 4). SSR consistently reduces the Harmfulness Score to near zero across all diverse safety evaluation datasets. This confirms that SSR repairs the fundamental "harmful pathways" introduced during poisoning, rather than overfitting to specific task features.
> >
> > **Q6**: OR/SR rates for baselines (Appendix B)
> >
> > We have evaluated the Over-Refusal (OR) and Successful Refusal (SR) rates for all baseline methods on the **GSM8K** fine-tuning task to ensure a fair comparison. The results are summarized in the table below:
> >
> > **Table R2: Over-Refusal (OR) vs. Successful Refusal (SR) Comparison (GSM8K)**
> >
> > | **Method**                | **OR Rate (%) ↓** | **SR Rate (%) ↑** | **Assessment**                                   |
> > | ------------------------- | ----------------- | ----------------- | ------------------------------------------------ |
> > | **Original (Base Model)** | 2.8               | 76.8              | Reference                                        |
> > | **SSR (Ours)**            | **10.8**          | **80.9**          | **Effective Defense & Balanced**                 |
> > | **SafeDelta**             | 60.0              | 76.0              | **Severe Over-Defense** (Refuses benign queries) |
> > | **SafeInstr**             | 1.1               | 6.1               | **Defense Failed** (Accepts harmful queries)     |
> > | **Safe-LoRA**             | 0.2               | 2.7               | **Defense Failed**                               |
> > | **SafeMERGE**             | 0.7               | 5.0               | **Defense Failed**                               |
> >
> > - **OR (Over-Refusal):** Percentage of benign prompts wrongly rejected (Lower is better).
> > - **SR (Successful Refusal):** Percentage of harmful prompts correctly rejected (Higher is better).
> >
> > The data reveals three distinct behaviors:
> >
> > 1. **Defense Failure (Safe-LoRA, SafeMERGE, SafeInstr):** These methods exhibit extremely low OR rates (<1.2%), but this is misleading. Their correspondingly low SR rates (<7%) indicate that they **fail to refuse harmful prompts entirely**, effectively passing through almost all inputs regardless of safety.
> > 2. **Severe Over-Defense (SafeDelta):** SafeDelta achieves a decent SR (76%), but at a catastrophic cost to utility: its OR rate skyrockets to **60.0%**. This implies the model has become overly conservative, refusing a majority of benign user queries.
> > 3. **Optimal Balance (SSR):** SSR is the only method that achieves a **State-of-the-Art SR (80.9%)** surpassing even the original aligned model while maintaining a manageable OR (10.8%). This demonstrates that SSR effectively distinguishes between safe and unsafe inputs, whereas baselines either fail to defend or resort to blanket refusals.

---

### Official Review · Reviewer_RoiX · 2025-11-02

**Soundness:** 3
**Presentation:** 2
**Contribution:** 2
**Rating:** 6
**Confidence:** 2

**Summary:**

This paper proposes SSR (Surgical Safety Repair), a framework designed to restore safety alignment in fine-tuned LLMs without requiring access to the original base model or a separately safety-aligned model. SSR works by (1) diagnosing harmful behaviors, (2) locating harmful LoRA parameters, and (3) updating only those parameters via a targeted unlearning-and-steering process. The authors argue that this approach achieves high safety recovery while preserving the model’s original utility, outperforming prior safety restoration methods.

**Strengths:**

1. Practical and deployment
* Unlike many existing methods that assume access to both base and safe models, SSR operates on a single compromised model, which is realistic for industry and proprietary settings. The method is lightweight because it relies on LoRA modification instead of full-model retraining.
2. Strong empirical results
* Harmfulness score reduced from ~77% → ~2%.
* Downstream task performance is largely preserved.
* Works consistently across multiple models (Llama3, Qwen2, Gemma2) and tasks (GSM8K, SST-5, PubMedQA).
3. Conceptual novelty
* Instead of globally modifying weights, the method isolates and edits only harmful LoRA units, which is a more surgical and controlled form of safety repair.
* Uses a simple but effective dual-objective loss combining unlearning (gradient ascent) and safety steering (gradient descent).

**Weaknesses:**

1. Strong dependence on LoRA assumption
* SSR only works directly for LoRA-based fine-tuned models. SSR cannot be applied to fully fine-tuned models unless additional steps are introduced. The paper claims “general applicability,” but does not demonstrate it experimentally.
2. Diagnostic dataset construction cost
* Although the paper claims a 100-example dataset is sufficient, the dataset requires manual safe-response curation, which introduces human cost, cultural bias, and reproducibility challenges.

**Questions:**

* Does modifying only the harmful LoRA units interfere with other task-specific capabilities encoded in those same parameters?
* If the diagnostic dataset reflects a specific culture or safety norm, does the model become biased toward that norm after repair?
* How stable is the attribution process under prompt variation?

---

> ### Author Response · Authors · 2025-11-23
>
> We thank the reviewer for the constructive feedback and for the positive recognition of our method. We are happy to further clarify the applicability scope of SSR and the cost-benefit analysis of our diagnostic dataset.
>
> **Weakness 1**: Dependence on LoRA assumption
>
> While our initial submission focused on LoRA due to its **prevalence in user-side fine-tuning**, we agree that demonstrating applicability to fully fine-tuned models strengthens our contribution.
>
> To empirically address the reviewer's concern, we applied SSR directly to models that underwent **Full Parameter Fine-Tuning (FFT)** on the GSM8K, SST5, and PubMedQA datasets. As shown in the table below, SSR successfully repairs fully fine-tuned models. Remarkably, even with a direct adaptation of the algorithm without extensive hyperparameter tuning specific to FFT, SSR achieves a massive reduction in harmfulness while preserving downstream capability.
>
> Table R1: SSR Performance on Fully Fine-Tuned Models
>
> (HS: Harmfulness Score, lower is better; FA: Fine-tuning Accuracy, higher is better)
>
> | **Task**     | **Method**     | **HS (%) ↓** | **FA (%) ↑** | **Result Analysis**                                     |
> | ------------ | -------------- | ------------ | ------------ | ------------------------------------------------------- |
> | **GSM8K**    | SFT (Full)     | 84.5         | 60.1         | Unsafe Baseline                                         |
> |              | **SSR (Ours)** | **11.8**     | **58.0**     | **Safety Restored ($\Delta$-72.7%), Utility Preserved** |
> | **SST5**     | SFT (Full)     | 81.4         | 57.6         | Unsafe Baseline                                         |
> |              | **SSR (Ours)** | **19.1**     | **57.5**     | **Safety Restored ($\Delta$-62.3%), Utility Preserved** |
> | **PubMedQA** | SFT (Full)     | 83.7         | 76.7         | Unsafe Baseline                                         |
> |              | **SSR (Ours)** | **12.9**     | **76.1**     | **Safety Restored ($\Delta$-70.8%), Utility Preserved** |
>
> The results demonstrate that the "Attribution-then-Editing" paradigm of SSR is optimization-agnostic:
>
> - **Significant Safety Gain:** SSR reduced the Harmfulness Score (HS) from saturation levels (>80%) to a safe range (11%–19%) across all tasks.
> - **Minimal Utility Cost:** The impact on downstream performance (FA) is negligible (e.g., only **-0.1%** drop on SST5 and **-0.6%** on PubMedQA), confirming that our parameter isolation strategy effectively targets harmful regions without disrupting the global knowledge structure of fully fine-tuned models.
>
> These new experimental findings provide evidence that SSR’s effectiveness may extend beyond its direct application with LoRA, addressing prior concerns on this matter. The core mechanism, which involves identifying and steering harmful parameters, is universally applicable. Whether the parameters come from a low-rank adapter or the full weight matrix, SSR effectively diagnoses and repairs the safety alignment.
>
>
>
> **Weakness 2**: Diagnostic dataset construction cost
>
> We believe the modest cost of constructing the diagnostic dataset is offset by its substantial benefits:
>
> 1. **Abundant Data Source & Minimal Overhead:** The construction cost is modest because **diverse harmful prompts are widely available in existing open-source benchmarks** (e.g., BeaverTails, HH-RLHF, as cited in Sec 3.2). Consequently, the human effort is strictly limited to curating **safe responses** for a small subset (100 examples), rather than inventing attacks from scratch. This is a one-time effort of a few hours—far less expensive  than curating the thousands of training samples required by baselines like SafeInstr.
> 2. **Human-AI Collaboration:** As detailed in Section 3.2, we employed a "Human-in-the-loop" process where powerful LLMs (e.g., Deepseek) generated initial drafts. Humans only verified and refined them. This ensures high quality with minimal manual labor.
> 3. **Reusability:** Once constructed, this small dataset serves as a "probe" that can be reused to diagnose and repair varied distinct models without modification, further spreading the cost.

---

> > ### Author Response · Authors · 2025-11-23
> >
> > **Question 1**: Interference with task-specific capabilities
> >
> > We acknowledge that modifying any parameters inevitably carries a risk of influencing other capabilities encoded in them. This is a **fundamental trade-off**: to effectively "unlearn" harmful behaviors, the model's internal representations must be altered. Without such modification, safety restoration is impossible.
> >
> > However, our **Parameter Isolation** mechanism is specifically designed to **minimize this collateral damage** while achieving the necessary safety correction:
> >
> > 1. **Surgical Precision:** Instead of a global update which degrades general capabilities (as shown in our ablation study with $p=1.0$, Figure 2b), SSR targets only the sparse subset of parameters most responsible for harmful outputs.
> > 2. **Minimal Impact:** Our experiments confirm that this trade-off is well-managed. For instance, while SSR drastically reduces the Harmfulness Score (HS) to <5%, the drop in Fine-tuning Accuracy (FA) is minimal (e.g., GSM8K: 67.6% $\rightarrow$ 66.5%), effectively preserving the model's overall structure and utility.
> >
> >
> >
> > **Question 2**: Cultural bias and norms
> >
> > Yes, the model's behavior **will naturally steer towards** the norms reflected in the diagnostic dataset. We consider this explicit steerability a **key feature** rather than a bug:
> >
> > 1. **Steerable Safety:** Traditional safety alignment is often opaque. SSR allows developers to explicitly define *what* constitutes a safe response. By modifying the small diagnostic set, the model can **adapt to** different cultural values or legal regulations.
> > 2. **Correction:** While the repair process introduces the biases present in the diagnostic set, correcting bias in just 100 examples is far more manageable and transparent than auditing a massive, black-box training corpus.
> >
> >
> >
> > **Question 3**: Stability of attribution under prompt variation
> >
> > The attribution process is designed to be robust to prompt variation. As shown in Equation (1), the contribution score $S_{l,i}$ is calculated as an **expectation** over the entire diagnostic dataset ($\mathbb{D}_{repair}$), not a single prompt. This averaging effect smooths out noise from individual prompt variations.

---

### Note · Authors · 2025-12-06

I have read and agree with the venue's withdrawal policy on behalf of myself and my co-authors.